# Experimental Study on Mechanical Properties of Root–Soil Composite Reinforced by MICP

**DOI:** 10.3390/ma15103586

**Published:** 2022-05-17

**Authors:** Xuegui Zheng, Xinyu Lu, Min Zhou, Wei Huang, Zhitao Zhong, Xuheng Wu, Baoyun Zhao

**Affiliations:** 1Science and Technology Department, Chongqing Vocational Institute of Engineering, Chongqing 402260, China; zheng_xuegui@126.com; 2School of Civil Engineering and Architecture, Chongqing University of Science and Technology, Chongqing 401331, China; 2020206077@cqust.edu.cn (X.L.); 2019441638@cqust.edu.cn (M.Z.); 2018445156@cqust.edu.cn (Z.Z.); 2019206040@cqust.edu.cn (X.W.); baoyun666@cqust.edu.cn (B.Z.)

**Keywords:** microbial reinforcement, root–soil composite, direct shear tests

## Abstract

Mechanical properties of undisturbed root–soil composites were investigated through direct shear tests under different cementation concentrations by microbially induced carbonate precipitation (MICP). The results show that MICP has a significant strengthening effect on the undisturbed root–soil composite, and the maximum shear strength increases by about 160% after grouting. The shear strength of root–soil composites increases with the increase in calcium chloride concentration, and the shear strength increases the most when the concentration is 0.75M. Calcium carbonate formed by MICP treatment has cementitious properties, which increases the cohesion and internal friction angle of the root–soil composite by about 400% and 120%, respectively. The results show that it is feasible to solidify slope and control soil erosion together with microbial and vegetation roots. The research results can serve as a scientific basis and reference for the application of MICP technology in vegetation slope protection engineering.

## 1. Introduction

In recent years, highway infrastructure construction has excavated vegetation mountains, which destroys the ecological environment and forms a large number of exposed slopes. How to restore the ecological environment and strengthen the slope is an urgent problem to be resolved in highway construction. The method usually used in current projects is vegetation slope protection. Its mechanism is that vegetation roots interpenetrate, entangle, and consolidate soil, forming root–soil composites similar to anchorage and reinforcement material, which plays a role in reinforcing rock and soil. The direct shear test [1] and the model box test [2] of the root–soil composite show that the shear strength of the reinforced soil is significantly improved. In addition, the root content in the root–soil composite is key to improving the strength of rock and soil [3], which is mainly achieved by increasing the cohesion between soil [4]. The vegetation slope protection method has superior environmental compatibility, and the slope protection effect is achieved by stabilizing the slope surface soil through the root–soil composite. However, in the actual projects, the initial vegetation root system is not developed, and the mechanical properties of the root–soil composite are poor, resulting in the inability of vegetation to protect slopes. Therefore, it is of great significance to seek new reinforcement methods for slope ecological governance.

Microbially induced carbonate precipitation (MICP) is an environmentally friendly microbial reinforcement technology. The technology uses urease produced by microorganisms for decomposing urea to generate carbonate ions, which combine with calcium ions to generate calcium carbonate with cementing ability to achieve the effect of biological reinforcement. At present, MICP technology has been proven to have significant effects in improving soil strength, stiffness, permeability, liquefaction resistance, and other related engineering problems [5,6,7,8,9,10]. Especially in the aspect of biological soil consolidation, the unconfined compressive strength of soil [11,12,13,14] and the slope stability coefficient are significantly improved after consolidation [15,16]. Meanwhile, MICP technology can restore slope ecology, because the cementation solution mainly contains urea, which can promote the rapid growth of the root system and strengthen the root–soil composite in the early stage of vegetation planting [17].

In conclusion, the combination of MICP technology and vegetation slope protection technology is a new method of ecological slope restoration and reinforcement, but its feasibility and effect lack corresponding research. Therefore, based on a slope project on Binjiang Road, Beibei District, Chongqing, China, in this paper, the effect of cementation concentration on the mechanical properties of the root–soil composites was studied, and experimental parameters were provided for the ecological restoration and reinforcement of the slope by MICP technology.

## 2. Materials and Methods

### 2.1. Root–Soil Composite

The sampling site of the root–soil composite is a soil slope on Binjiang Road, Beibei District, Chongqing, China, which is protected by vegetation with bermudagrass. When sampling, a slope with the same density of bermudagrass plants was selected, a root soil drill with an inner diameter of 61.8 mm was used to obtain the undisturbed root–soil composite, and the soil was cut on both sides of the ring knife to the level to obtain natural roots for the direct shear test. There are 15 soil complexes in total, and the sampling process is shown in Figure 1.

According to the geotechnical test method standard (GBT 50123-2019), the physical and mechanical indicators of the undisturbed root–soil composite are shown in Table 1.

### 2.2. Bacteria and Growth Medium

The selected strain is Sporosarcina pasteurii in this test, which was purchased from the China General Microbiological Culture Collection Center, Beijing, China. The CGMCC number is 1.367, and this strain is an enzyme-producing microorganism and a non-pathogenic strain. Each liter of bacterial culture solution contains 20 g of yeast extract, 10 g of NH_4_Cl, 20 mL of Ni(Cl)_2_·6H_2_O with a concentration of 2.4 g/L, and 20 mL of MnSO_4_·H_2_O with a concentration of 1 g/L. Then, slowly sodium hydroxide solution was added until the pH of the solution stabilized at around 9. The activated bacterial solution was inoculated into the plate and placed in a 30 °C microbial constant temperature incubator for 2–3 days. After the white colonies grew on the plate, they were stored in a 4 °C refrigerator.

When selecting colonies, after sterilizing the inoculation loop with an alcohol lamp, it was inoculated into 100 mL of culture medium and placed in a constant temperature shaking incubator for 24 h at 30 °C and 200 r/min, and then a conductivity meter was used to measure the bacterial urease activity. In this experiment, 2 mL of bacterial solution was mixed with 18 mL of 1.1 M urea solution, and the conductivity change in the mixed solution was measured by a conductivity meter at room temperature within 5 min, and the average conductivity change value per minute (mS1·min^−1^). According to the empirical formula of the relationship between urea hydrolysis amount and conductivity change obtained by Whiffin [19], the ability of bacteria to hydrolyze urea per unit time (i.e., urease activity) is characterized. The urease activity of the bacterial solution used in this experiment was 1.1 mM urea hydrolysed·min^−1^.

### 2.3. Cementation Concentration

The cementation concentration used for grouting are calcium chloride (CaCl_2_) and urea in this test. The cementation concentrations and numbers are shown in Table 2. In the process of microbial reinforcement reaction, urea provides a nitrogen source and energy source for the growth of microorganisms, and CaCl_2_ provides a calcium source for the reaction.

### 2.4. Sample Preparation

Due to the small size of the sample, in order to prevent disturbance of the undisturbed soil, straight liquid pinhole grouting was adopted in this test. The ratio of injection of bacterial solution and cementation solution was 1:1, and each injection was 4 mL each time. In order to ensure uniformity of the sample grouting, the three-hole grouting method was adopted to make the bacterial solution and the cementation solution penetrate evenly. On the second day, the sample was grouted by flipping over to achieve double-sided penetration and curing and ensure the cementation effect of the sample. The grouting position is shown in Figure 2. During the grouting process, in order to prevent the injection solution from overflowing, the samples were wrapped with plastic wrap. After grouting, they were placed in a greenhouse with an indoor temperature of 30 °C to ensure the activity of bacteria. The 0.00 M control sample was replaced with an equal volume of distilled water instead of bacterial solution and cementation solution, while the other steps remained unchanged. After the grouting process of the sample was completed, it was placed in the curing tank for 7 d to ensure the complete reaction between the bacterial solution and cementation solution. Grouting was carried out twice a day, the interval time was 12 h, and the total reaction time was 10 d.

### 2.5. Experimental Instruments

In this test, pH was measured by PHS-3C pH meter (for China), bacterial activity was measured by DDS-11A conductivity meter (for China). The ZJ-3 strain-controlled quadruple direct shear instrument (for China) was used to conduct a quick shear test on the MICP-reinforced root–soil composite. During the test, 100 kPa, 200 kPa, and 300 kPa were selected as the vertical pressure, the shear rate was controlled at 0.8 mm/min, and the shearing was stopped at 6 mm. SEM was measured by KYKY-EM6200 SEM analyzer (for China).

## 3. Results

MICP reinforces the undisturbed root–soil composite. Its mechanism is that the metabolism of Sporosarcina pasteurii produces urease, which hydrolyzes urea to generate NH_4_^+^ and CO_3_^2−^. The surface of cells is negatively charged, and when Ca^2+^ is present in solution, the positively charged calcium ions will be absorbed, resulting in calcium carbonate precipitation. The reaction mechanism is shown in Figure 3. Calcium carbonate fills in the pores of the root–soil composite and plays a role in cementing and hardening, thereby increasing the shear strength of the soil.

### 3.1. Shear Strength Analysis

The variation curve of shear strength with calcium ion concentration is shown in Figure 4. It can be seen from the figure that the shear strength increases with the increase in calcium ion concentration, and its growth rate was a maximum at 0.75 M concentration. This phenomenon is consistent with the conclusion of Wu et al. [11] in the experiment of microbial reinforcement of sand: the higher the cement concentration, the higher the contribution of the unit CaCO_3_ content of the microbially cured sample to the peak strength, but the lower the CaCO_3_ content produced by the consumption of the unit cement concentration. In this study, although the strength is the highest when the cementation concentration is the highest, the strength growth rate is the highest at 0.75 M, that is, the strength increase efficiency is the highest at this time. This is due to the inhibitory effect of urease in high concentrations of the cementation solution. When the cementation concentration reaches a certain value, the concentration of carbonate ions generated by urease hydrolysis decreases, resulting in a decrease in the amount of CaCO_3_ precipitation, so the cementation tends to decrease. On the other hand, some studies have shown that the permeability of soil decreases significantly with the increase in cementation level [12,20]. The CO_3_^2−^ produced by urea under the catalysis of urease secreted by bacteria and the artificially added Ca^2+^ generate CaCO_3_ which precipitates in the soil, fills the soil pores, and the permeability decreases. High concentration led to a quick response to focus on grouting at the surface, leading to different permeability of sample, and the uneven distribution of CaCO_3_ precipitation in the soil [13,21] has significant effects on the cementation strength of the sample.

At the same time, the test results also show that after grouting, the soil pores are filled with calcium carbonate cementation, resulting in increased flow resistance and a new preferred liquid flow path [22]. In addition, the new preferred flow path would bring bacteria and cementation solution to areas with less precipitation, so that the precipitation is evenly distributed in the soil, and the cementation strength is improved.

In order to obtain cohesion (*c*) and internal friction angle (*φ*), the shear stress under different vertical stresses was fitted by a straight line. The intersection of the fitted line with the vertical axis is *c*, and the angle with the horizontal axis is *φ*. In order to simplify the expression, the experimental curve of 0.00 M concentration was selected for fitting, as shown in Figure 5.

By fitting the shear stress under different vertical pressure, the shear strength parameters *c* and *φ* of different concentrations were obtained, as shown in Table 3. With the increase in the cementation concentration, the values of *c* and *φ* increased approximately linearly, and Canakci [23] also drew the same conclusion. The reason is that the calcium carbonate precipitates generated in the MICP process fill the pores of soil and form an effective connection point to connect loose soil particles, which improves the cohesion of the root–soil composite. In addition, calcium carbonate crystals coated the soil particles, changed the particle size and surface roughness of the original soil, and increased the internal friction angle.

Further analysis shows that compared with the control group (0.00 M), the increase in cohesion after 1.00 M reinforcement is 400% and the increase in internal friction angle is 125%. The increase in cohesion is much larger than that of the internal friction angle, which is contrary to the result of Shen [24]. Shen studied the microbial reinforcement effect of sandy cohesive purple soil, and its cohesion and internal friction angle increased by 120% and 300%, respectively. The main reason is that since the root–soil complex is relatively plain soil, the root system provides the additional cohesive force, after MICP reinforcement, so that the effective connection points formed by calcium carbonate cementation between soil particles have a significant contribution to the cohesion. That is, increase in cementation concentration will increase the production of calcium carbonate. Therefore, the cohesion of the root–soil composite reinforced by MICP plays a dominant role in the improvement of shear strength.

### 3.2. Deformation Failure Analysis

Different cementation concentrations will cause different cementation levels of an undisturbed root–soil composite, so that its shear strength is also different. This study is in accordance with the standard “Geotechnical Test Method Standard” (GB/T 50123-2019). The shear displacement and shear stress curves under the vertical pressure of 100 kPa, 200 kPa, and 300 kPa are shown in Figure 6.

As can be seen from the figure above, the shear stress curve shows strain hardening, and with the gradual the increase in cementation concentration, the shear stress gradually increases. According to “Standard for Geotechnical Test Methods” (GB/T 50123-2019), shear stress corresponding to shear displacement L = 4 mm is taken as shear strength. The shear strength of the control sample (the cementation concentration of 0.00 M) is 60.8 kPa, and the strength growth rate after 0.25–1.00 M is about 10%, 30%, 50%, and 60%, respectively. Compared to microbial reinforced clay, the 1 M sample (with similar basic physical properties of plain soil) studied by Peng [13] only increased by 8.9%. The reason is that the shear strength of the root–soil composite is composed of plain soil strength and additional root cohesion. When the strength of plain soil is similar, additional cohesion generated by root tension increases the shear strength of the root–soil composite.

Under the same shear stress, the shear displacement after reinforcement (1.00 M) was much smaller than that generated by the control group (0.00 M), and the sample showed a large amount of white calcium carbonate precipitation after reinforcement (Figure 7). Studies have shown that there is a positive correlation between the content of calcium carbonate in soil samples and soil strength, and the amount of calcium carbonate generated and the content of calcium ions in the cementation solution are both positively correlated [25,26,27]. The only variable of the cementation concentration in this test is the content of CaCl_2_. Based on the test results, it is initially effective to improve the cementation level of the root–soil composite by increasing the content of calcium ions in the grouting solution, and the shear strength of the sample reached the maximum when urea and CaCl_2_ were both 1 mol/L.

### 3.3. SEM Analysis

The microstructure of the MICP-reinforced soil was observed by SEM. Figure 8 and Figure 9 are the SEM microstructure images of soil samples before and after MICP treatment, respectively.

It can be seen that the soil particles that have not been treated with MICP are flocculent, there are many pores in the soil, and the structure is relatively loose. However, the soil samples treated with MICP include diamond-shaped calcites. These calcium carbonates produce cementation in the soil, which increases the cohesion of the soil, enhances the internal stability of the soil, and gives the soil better mechanical properties. During the MICP reaction, the calcium carbonate crystals accumulated continuously, forming calcium carbonate particle clusters on the surface of soil particles. The addition of roots established a bridge between soil particles effectively. Reinforcing root–soil composites with MICP has a remarkable effect.

## 4. Conclusions

In this paper, Bacillus pasteuri was used to reinforce the undisturbed root–soil complex in vegetation slope protection by MICP. Based on the direct shear test, by controlling the calcium ion concentration in the cementation solution, the mechanical properties of the root–soil composite with different cementation concentrations were explored. The following conclusions are drawn:

(1)MICP can improve the shear strength of undisturbed root–soil composites. Compared with the solidified ordinary cohesive soil, the strength increase is 110–140%, and the root system in the root–soil composite has a positive effect on the increase in its shear strength.(2)The higher the concentration of CaCl_2_ in the cementation solution, the more obvious the improvement of the shear strength of the root–soil composite, and the high concentration of calcium ions will promote the reaction and produce more calcium carbonate precipitation. Under the condition of constant urea and root content, when the concentration of CaCl_2_ in the cementation solution is 1 mol/L, the solidification effect of the root–soil composite is the best, but the increase in strength is the largest when the concentration is 0.75 mol/L.(3)After microbial reinforcement of the root–soil composite, the increase in cohesion is greater than that of the internal friction angle. In practical engineering applications, the strength parameters of the root–soil composite can be effectively improved by increasing the amount of calcium carbonate precipitation.

## Figures and Tables

**Figure 1 materials-15-03586-f001:**
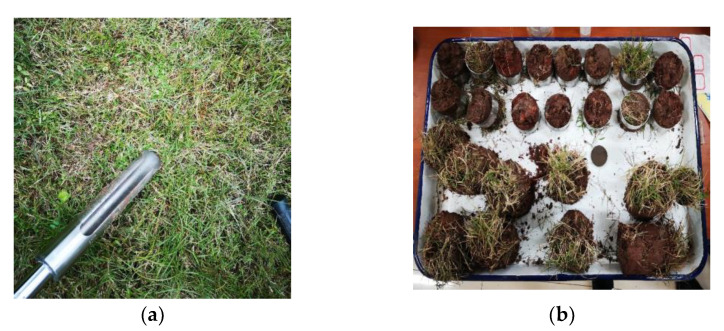
Root–soil composite sample. (**a**) root soil drill; (**b**) undisturbed root–soil composite.

**Figure 2 materials-15-03586-f002:**
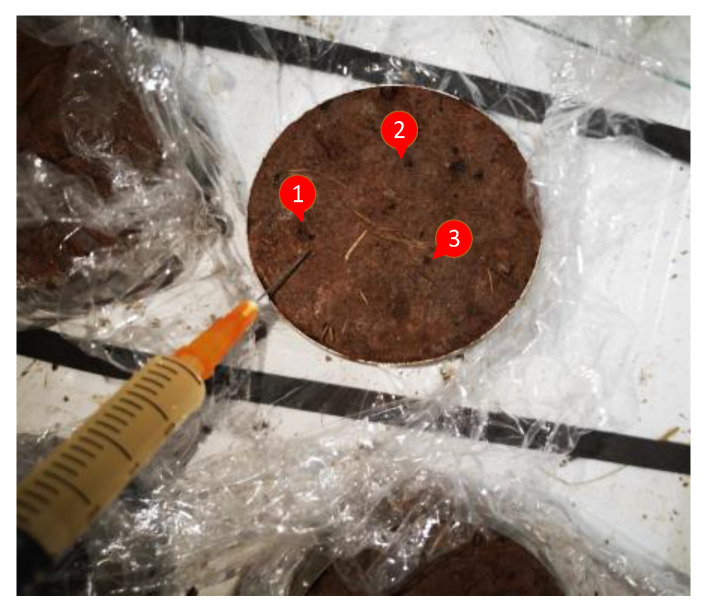
Schematic diagram of undisturbed root–soil composite grouting. (The numbers 1, 2 and 3 indicate the injection location in the figure.)

**Figure 3 materials-15-03586-f003:**
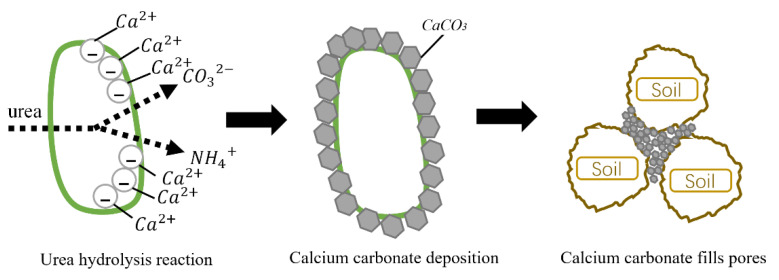
Microbially induced calcite precipitation (MICP) mechanism.

**Figure 4 materials-15-03586-f004:**
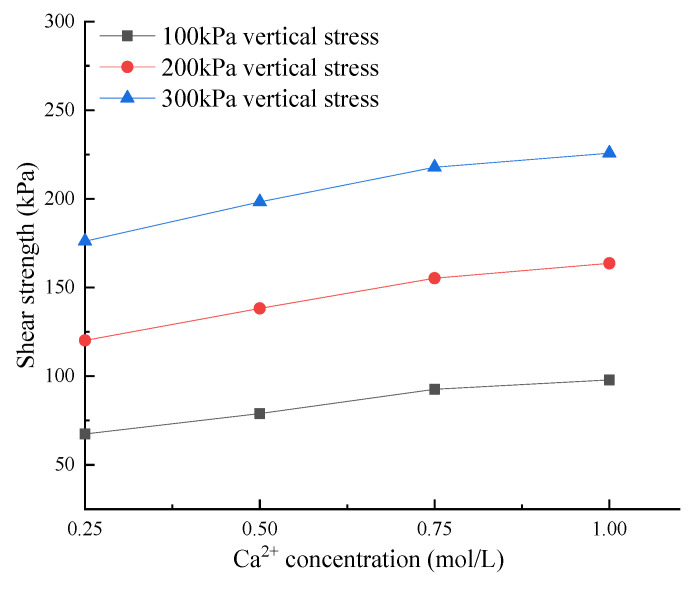
Shear stress variation with calcium ion concentration.

**Figure 5 materials-15-03586-f005:**
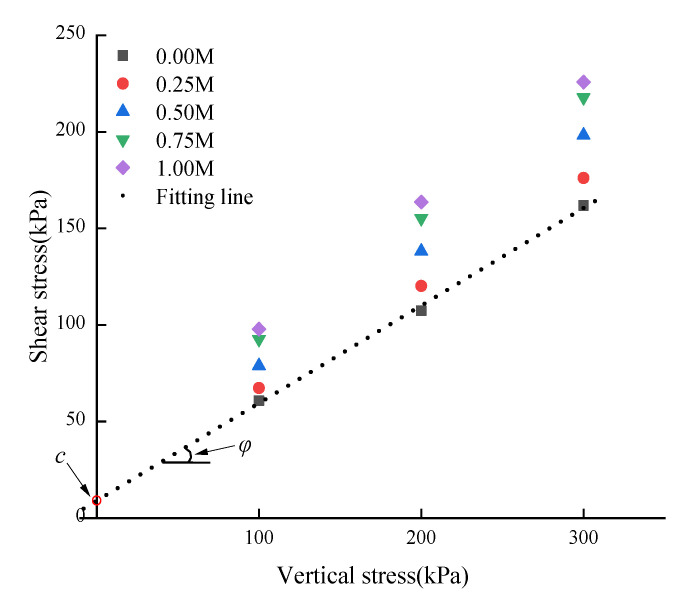
Shear stress variation with vertical stress.

**Figure 6 materials-15-03586-f006:**
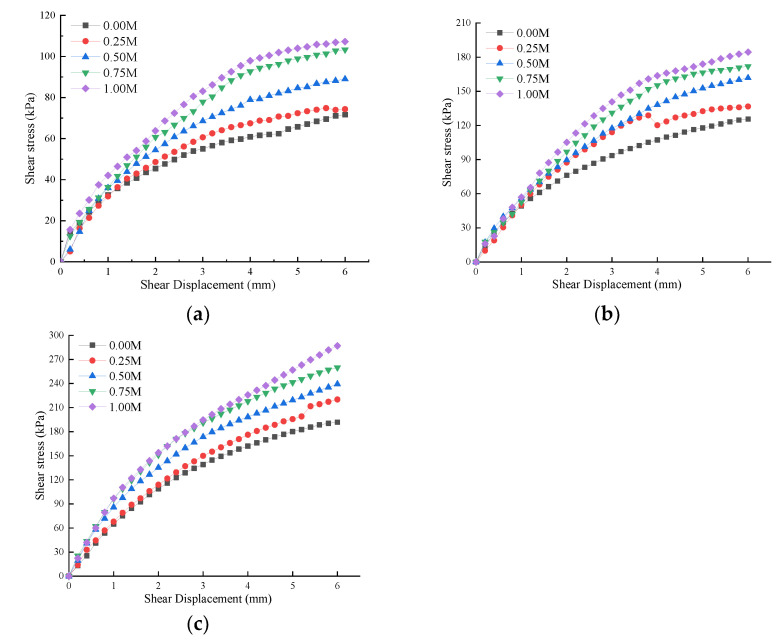
Shear stress and displacement curve: (**a**) 100 kPa; (**b**) 200 kPa; (**c**) 300 kPa.

**Figure 7 materials-15-03586-f007:**
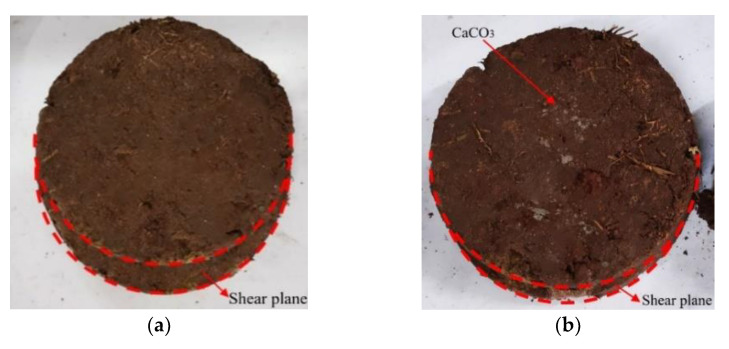
Shear plane of samples under different cementation concentrations. (**a**) 0.00 M sample; (**b**) 1.00 M sample.

**Figure 8 materials-15-03586-f008:**
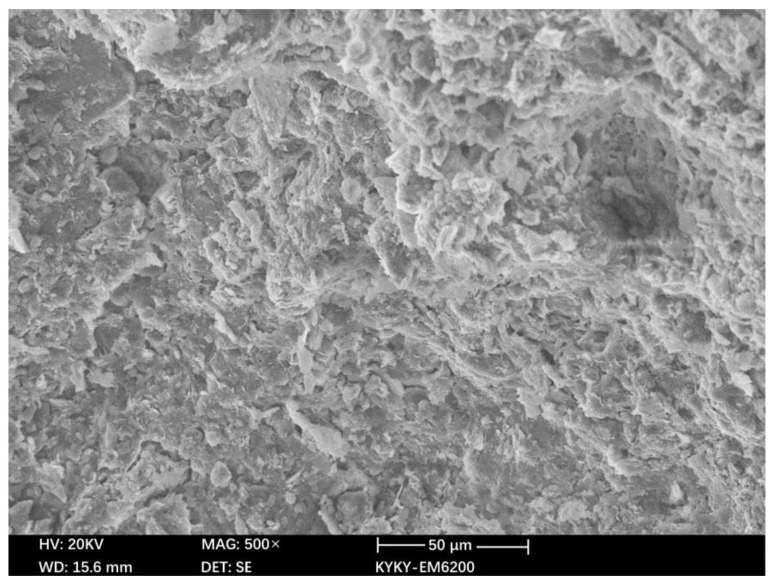
SEM microstructure of soil samples before MICP treatment.

**Figure 9 materials-15-03586-f009:**
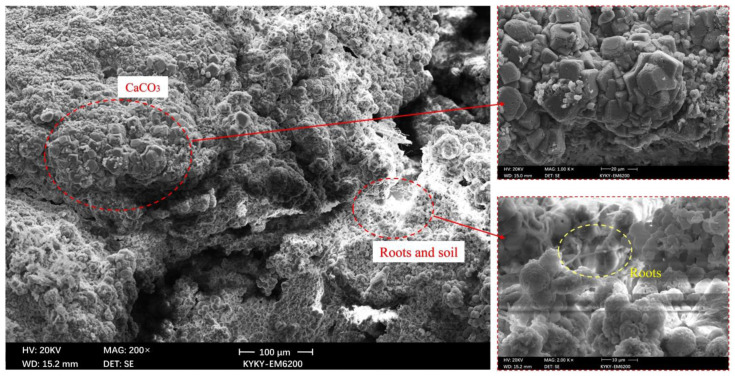
SEM microstructure of soil samples after MICP treatment.

**Table 1 materials-15-03586-t001:** Basic physical and mechanical properties of soil [18].

Properties	Values
Density, *ρ* (g/cm^3^)	1.91
Moisture content, *w* (%)	22
Specific gravity, *d*_s_	2.56
Plastic limit, *w*_P_ (%)	19.2
Liquid limit, *w*_L_ (%)	33.2
Plasticity index, *I*_P_ (%)	14
Liquidity index, *I*_L_	0.2

*w*_P_: refers to the limit water content from a plastic state to a semi-solid state; *w*_L_: refers to the limit water content of soil from flowing state to plastic state; *I*_P_: the difference between the liquid limit and the plastic limit is called the plasticity index. Plasticity is an important feature to characterize the physical properties of fine-grained soil, and is generally expressed by plasticity index; *I*_L_: an index that expresses the relative relationship between natural water content and limit water content, and it is an index for judging the soft and hard state of soil.

**Table 2 materials-15-03586-t002:** Sample number and cementation concentration.

Concentration Number	CaCl_2_ Concentration /(mol L^−1^)	Urea Concentration /(mol L^−1^)	Number of Samples
0.00 M	0	0	3
0.25 M	0.25	1	3
0.50 M	0.50	1	3
0.75 M	0.75	1	3
1.00 M	1.00	1	3

**Table 3 materials-15-03586-t003:** Mechanical parameters of shear strength.

Concentration Number	*c*/(kPa)	*φ*/(°)
0.00 M	8.9	26.8
0.25 M	12.5	28.5
0.50 M	19.1	30.8
0.75 M	30.1	32.0
1.00 M	34.5	32.6

## Data Availability

All the data in the tests of this study have been listed in the paper.

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
