# Peer review of "Experimental Study on Mechanical Properties of Root–Soil Composite Reinforced by MICP"

_materials, 2022, doi:10.3390/ma15103586_

Round 1
Reviewer 1 Report
Good work, please have a look on the attached file for improvement purposes. I have marked some Important comments needed to be addressed prior to the acceptance

Author Response
请参阅附件。

Reviewer 2 Report
Review for materials-1628758
My review is centered around a scanned copy of the manuscript with handwritten notes that might help improve the language. There are also a lot of question marks in particular versus the end of the manuscript, where I did not have the ambition any more to find potential corrections. Sentences seemed incomplete and not understandable. I would encourage the authors to have this read by a native speaker so that the text becomes more readable at those passages.
The scanned copy also contains numbered items #i to which I will refer in the following.
#1 here, and later I think the authors should pay attention to digits. i.e. is it really necessary to give the .7 here? And is it justified?
#2 here it was not clear where the concentration came from. Maybe earlier something should be said about the experiment, even in line 11 the cementation concentration would not be clearly understandable.
#3 I think the authors want to say “inability” here instead of “ability”. To me otherwise it makes no sense. Please check.
#4 For several of the quantities in Table 1 the units seem to be missing.
#5 The quantities in Table 1 are not explained at all. Please add definitions of those that are not generally known (i.e. plastic limit, liquid limit, plastic index and liquidity index). I would even prefer that you give physical symbols to all those quantities and define them in the text or in a footnote to the table.
#6 I think the authors want to say that the bacterial urease activity converted urea to something… Please check.
#7 There is a lot of information missing in the experimental part:
- How was the pH measured? Set-up, calibration, temperature.
- How was the conductivity measured? Set-up, calibration, temperature.
- The CaCl2 and urea chemicals need to be specified. Were stock solutions prepared? What was their pH and concentration?
- At the bottom of the page in inserted a section 2.5, where the used instruments (e.g. the SEM, the shear instrument etc.) need to be specified.
#8 I see only one sample number in Table 2. Was there only one and if so, why do you give it a number then? If there were several, why do you not report on them here.
#9 This section should be part of the experimental procedures.
#10 it is not clear what the authors want to say. I think they refer to the conclusion of Wu et al. and do not give their own argument.
#11 This sentence is not clear to me at all.
#12 in table 3, what are the significant digits?
#13 again, I do not seem to remember here a clear description of how the values were determined. And I miss the definition of the internal friction angle for example. Same as in #5, one might define physical symbols, define what they are and how they are measured.
#14 This is not clear to me at all. Figure 5 shows curves at different concentrations. I do not see curves at different vertical pressures? Why are they not shown, and if there is more material than what the authors want to show in the main text, why not give all the data and put them in supplementary information?
# 15 At this point I started wondering about the the optimum mentioned in the abstract at 0.75 M. It seems to me that this is only mentioned in line 135, but the results are not really shown. All results that are actually shown rather show that 1 M is giving the best results.
#16 in general, since nothing is said about solution preparation and so on, I was wondering about details, also because Figure 3 suggests that carbonate is coming from urea. However, carbonate is also from the environment (from carbon dioxide) and formation of calcite can also occur in highly concentrated Ca-solutions depending on the pH for example. Therefore, it seems necessary to specify the conditions. This concerns what was the pCO2 in the experiments and for example to say whether the pH remained constant or not.
In general, I like the work and think it is important. However, due to the lack of information and some shortcomings in the writing etc. I would go for major, major revisions.

Round 2
Reviewer 1 Report
Agreed, Thanks for considering my comments positively